# HBCR_DMR: A Hybrid Method Based on Beta-Binomial Bayesian Hierarchical Model and Combination of Ranking Method to Detect Differential Methylation Regions in Bisulfite Sequencing Data

**DOI:** 10.3390/jpm14040361

**Published:** 2024-03-29

**Authors:** Maryam Yassi, Ehsan Shams Davodly, Saeedeh Hajebi Khaniki, Mohammad Amin Kerachian

**Affiliations:** 1Cancer Genetics Research Unit, Reza Radiotherapy and Oncology Center, Mashhad 9184156815, Iran; maryam.yassi@postgrad.otago.ac.nz (M.Y.); ehsan_shams_davodly@alumni.um.ac.ir (E.S.D.); 2Department of Mathematics and Statistics, University of Otago, Dunedin 9054, New Zealand; 3Department of Pathology, Dunedin School of Medicine, University of Otago, Dunedin 9054, New Zealand; 4Student Research Committee, Department of Biostatistics, School of Health, Mashhad University of Medical Sciences, Mashhad 9177948564, Iran; hajebis971@mums.ac.ir; 5Medical Genetics Research Center, Mashhad University of Medical Sciences, Mashhad 9177948564, Iran; 6Department of Medical Genetics, Faculty of Medicine, Mashhad University of Medical Sciences, Mashhad 9177948564, Iran; 7Department of Chemistry and Biology, Toronto Metropolitan University, Toronto, ON M5B 2K3, Canada

**Keywords:** DNA methylation, epigenetic, differentially methylated region, beta-binomial Bayesian hierarchical model, ranking method

## Abstract

DNA methylation is a key epigenetic modification involved in gene regulation, contributing to both physiological and pathological conditions. For a more profound comprehension, it is essential to conduct a precise comparison of DNA methylation patterns between sample groups that represent distinct statuses. Analysis of differentially methylated regions (DMRs) using computational approaches can help uncover the precise relationships between these phenomena. This paper describes a hybrid model that combines the beta-binomial Bayesian hierarchical model with a combination of ranking methods known as HBCR_DMR. During the initial phase, we model the actual methylation proportions of the CpG sites (CpGs) within the replicates. This modeling is achieved through beta-binomial distribution, with parameters set by a group mean and a dispersion parameter. During the second stage, we establish the selection of distinguishing CpG sites based on their methylation status, employing multiple ranking techniques. Finally, we combine the ranking lists of differentially methylated CpG sites through a voting system. Our analyses, encompassing simulations and real data, reveal outstanding performance metrics, including a sensitivity of 0.72, specificity of 0.89, and an F1 score of 0.76, yielding an overall accuracy of 0.82 and an AUC of 0.94. These findings underscore HBCR_DMR’s robust capacity to distinguish methylated regions, confirming its utility as a valuable tool for DNA methylation analysis.

## 1. Introduction

Epigenetics is a research area that offers insight into the activation or suppression of genes within living cells, revealing the how, where, and when of these processes. DNA methylation has been extensively researched and is well understood as an epigenetic mechanism that plays a crucial role in various processes [1], including cell development and differentiation. DNA methylation patterns, characterized by either hypo- or hypermethylation, have been identified in human tumor cells, offering valuable insights into the development and progression of complex diseases [2].

Methylation is a process in which methyl groups are added to DNA cytosine (C) molecules, typically occurring at CpG sites. Methylation of promoter regions is commonly associated with gene expression suppression, whereas methylation within gene bodies is generally linked to increased gene expression [3].

In brief, when DNA is treated with bisulfite, unmethylated cytosines are converted to uracil (U), while methylated cytosines remain unchanged. Sequencing of bisulfite-treated DNA and aligning the sequenced reads to a reference genome allows for the quantification of methylation levels at each cytosine. Methylation can occur in three different sequence contexts: CpG, CHG, and CHH (where H corresponds to A, T, or C). Additionally, CHG and CHH methylation has been reported on rare occasions [4]. In this discussion, we will focus solely on the methylation of individual cytosine nucleotides.

Whole-genome bisulfite sequencing (WGBS) enables the precise measurement of DNA methylation across the entire genome [5]. However, alternative DNA methylation sequencing methods have been developed to cost-effectively cover variable regions of DNA methylation. These methods often employ a reduced representation of bisulfite sequencing, focusing on specific restriction sites, such as Reduced Representation Bisulfite Sequencing (RRBS) [6].

One of the most reliable and widely adopted approaches for measuring DNA methylation is the SureSelectXT Human Methyl-seq method. This platform evaluates 84 megabases (MB) of the genome, encompassing 3.7 million CpGs, 19.6 megabases of CpG islands, 9.8 megabases of cancer- and tissue-specific DMRs, 37 megabases of GENCODE promoters, 48 megabases of enhancers, CpG island shores/shelves within ±4 kilobases, and DNase I hypersensitive sites [7].

The epigenetic differences between sample groups are typically described by differentially methylated cytosines (DMCs) and differentially methylated regions (DMRs). The DNA methylation sequencing data comprise three steps in the pre-processing stage before detecting DMRs. Firstly, the total reads are assessed by a Quality Control (QC) tool to provide informative global and graphical representations of methylation sequencing read quality, which are typically applied both before and after alignment (Wingett and Andrews 2018).

Secondly, the unprocessed sequencing reads undergo cleaning through Trim Galore (https://www.bioinformatics.babraham.ac.uk/projects/trim_galore/, accessed on 8 September 2023), which involves the removal of sequencing adapters (specifically the Illumina universal adapter), elimination of low-quality bases (those with Q < 67 in Illumina) at the 3′ end, and handling of ambiguous bases in both reads. Thirdly, these initial bisulfite sequencing data are transformed into a count of methylated reads and covered reads of cytosines (comprising both unmethylated and methylated reads) by aligning them to the human reference genome.

For instance, conversion-aware aligners such as BSMAP [8], Bismark (Krueger and Andrews 2011), MethylCoder [9], BRAT-BW [10], Last [11], BS-Seeker2 [12], Bison [13], bwa-meth, WALT [14], VaiBS [15], BiSpark [16], BS-Seeker3 [17], and gemBS [18] are utilized to align the sequenced fragments of bisulfite-treated DNA. The methylation status of a CpG site is documented by counting the reads that are methylated and unmethylated, spanning each specific site. According to a comprehensive review article [19], differential methylation finder methods are classified into seven categories based on their primary concepts and features. For example, Logistic regression-based approaches like methylkit [20] and eDMR [21], Smoothing-based approaches like Bsmooth [22], Biseq [23], and HOME [24], Bata-binomial-based approaches like DSS [25], MOABS [26], RADMeth [27], methySig [28], DSS-signal [29], MACAU [30], DSS-general [30], and GetisDMR [31], Hidden Markov model-based approaches like ComMet [21], HMM-Fisher [32], HMM-DM [33], and DMCHMM [34], Entropy-based approaches like QDMR [35], CpG_MPs [36], and SMART [37], Mixed statistical test-based approaches like COHCAP [38], DMAP [39], and swDMR [40], and Binary segmentation-based approaches like metilene [41] and MethCP [42].

In the current paper, we introduce a novel DMR finder designed for the detection of DMRs in bisulfite sequencing data. The HBCR_DMR method is founded on a hybrid approach that combines two statistical methods: a beta-binomial Bayesian hierarchical model and a combination of ranking techniques. Within the HBCR_DMR method, we assess the variation across CpG methylation proportions using the beta-binomial model. Additionally, for DMR detection, we employ a combination of ranking methods to select discriminative CpG sites.

HBCR_DMR is versatile and can be employed with a range of methylation sequencing platforms, such as WGBS, RRBS, and target-capture methods. In this investigation, we employ HBCR_DMR for the examination of both simulated data (RRBS) and authentic data obtained from colorectal cancer samples (SureSelectXT Human Methyl-seq analysis).

## 2. Method

Our proposed method consists of six main stages, including (1) CpG clustering, (2) mean and variation assessment using a beta-binomial hierarchical model, (3) ranking method for identifying distinguishing CpG site selection based on methylation status, (4) combination of ranking methods, (5) definition of DMR boundaries, and (6) annotation/visualization. We identified the discriminating DMRs in simulation and real datasets and compared the selected DMRs found by HBCR_DMR with other methods. Figure 1 illustrates the flowchart of our proposed method. Our method is an open source software program and is available on GitHub (https://github.com/Genetics-Research-Laboratory-RROC/HBCR-DMR, accessed on 8 September 2023).

### 2.1. Data

To assess our method on simulation data, we utilized the “RRBSdata” R package, which includes twelve samples divided into two groups: six controls and six cases. The RRBSdata package encompasses a total of 7,986,265 CpGs, including 24,698 CpG islands. Additionally, it incorporates 10,000 simulated DMRs sourced from a previously published RRBS dataset with the accession number GSE42119 [43].

Furthermore, for real data analysis, we employed the SureSelectXT Human Methyl-Seq approach with a 101-base read length. This method generated 57–76 million Illumina sequencing reads from a dataset comprising six colon adenocarcinomas and six control samples (colon normal tissues). Remarkably, 88.5% to 89.8% of these reads were successfully mapped to either strand of the human genome (GRCh37/19). On average, each CpG was sequenced between 19X and 24X per sample (Appendix A).

### 2.2. CpG Clusters

Every CpG cluster signifies a region of the genome abundant in CpG sites. In each of these clusters, we filter out extraneous data noise from the complete genome’s DNA methylation dataset. This not only enhances data quality but also streamlines computational processing time by concentrating on defined and scrutinized genomic regions.

Discovering the start and end points of each CpG cluster consists of two steps:(1)CpG sites found in the majority of at least 75% of all samples are designated as validated CpGs. If the occurrence of any CpG site across all samples falls below the 75% threshold, it is categorized as “noise” and subsequently removed from the DNA methylation dataset [23].(2)A CpG cluster is defined as a collection of validated CpGs from all samples when the maximum distance between individual CpG sites within it is less than 100 base pairs.

### 2.3. Beta-Binomial Hierarchical Model

In our approach, we estimate both variation and mean values using a beta-binomial hierarchical model [25]. This model’s prior distribution is based on the entire genome, considering both methylated and unmethylated states. The genuine methylation proportions of CpGs within the replicates are represented by a beta distribution, parameterized by a group mean and a dispersion parameter. The beta distribution accounts for biological variability, while the binomial distribution captures sampling variability. To quantify variation in CpG methylation proportions concerning the group mean, we employ the dispersion parameter $\varphi_{ij}$, which is estimated through an empirical Bayes method. Detailed statistical formulas and notations are presented below:(1)$$\left(X_{ijk}|P_{ijk},N_{ijk}\right)\sim Binomial\;\left(N_{ijk},P_{ijk}\right)$$
(2)$$P_{ijk}\sim Beta(u_{ij},\varphi_{ij}),\;\mu =\frac{\alpha }{\alpha +\beta },\;\varphi =\frac{1}{\alpha +\beta +1}$$
(3)$$\varphi_{ij}\sim log-normal\;\left(m_{0j},r_{0j}^{2}\right)$$
(4)$${\hat{\mu }}_{ij}=\frac{\sum_{k}X_{ijk}}{\sum_{k}N_{ijk}},\;{\hat{var}}_{ij}={\left(\frac{1}{\sum_{k}N_{ijk}}\right)}^{2}\sum_{k}\left\{N_{ijk}u_{ij}(1-u_{ij})\left[1+(N_{ijk}-1)\varphi_{ij}\right]\right\}$$

In Equation (1), denoting the *i*-th CpG site, *j*-th group, and *k*-th replicate, we have $X_{ijk}$ as the count of reads indicating methylation, $N_{ijk}$ as the total number of reads covering this position, and $P_{ijk}$ as the true underlying methylation proportion.

In Equation (2), the beta distribution is parameterized with a mean (represented as $u_{ij}$) and a dispersion (denoted as $\varphi_{ij}$). Compared with the traditional parameterization of the beta (α,β) distribution. In Equation (3), we make the assumption that the dispersion parameters can be effectively described by a log-normal distribution with a mean of –3.39 and a standard deviation of 1.08. Equation (4) outlines how the mean methylation levels, denoted as ${\hat{\mu }}_{ij}$, are estimated, and how $\varphi_{ij}$ is obtained by maximizing the conditional posterior likelihood. Consequently, ${\hat{var}}_{ij}$ represents the estimated variance for the *j*-th group. For each CpG site within a specific condition (e.g., cases or controls), unique mean methylation levels ${\hat{\mu }}_{ij}$ and variances ${\hat{var}}_{ij}$ are estimated using the beta-binomial hierarchical model.

### 2.4. Ranking Method

Detecting differentially methylated regions (DMR) in bisulfite sequencing poses a significant challenge, primarily centered around the selection of CpG sites based on their methylation status. According to DMRFusion [44], CpG sites are typically chosen individually, without considering the interrelationship between features, using rankings based on the relative methylation levels in cancer and control groups. These rankings help determine the usefulness of each CpG site for DMR detection and employ selection methods such as Information gain, Between versus within Class scatter ratio, Fisher ratio, Z-score, and Welch’s *t*-test. HBCR_DMR uses three ranking methods—Fisher ratio, Z-score, and Welch’s *t*-test—which are based on the normal distribution. In our approach, as detailed in the previous section, we estimate mean methylation levels ${\hat{\mu }}_{ij}$ and variances ${\hat{var}}_{ij}$ for each CpG site using a beta-binomial hierarchical model for both control and normal groups. These estimates are then utilized in the Fisher ratio, Z-score, and Welch’s *t*-test ranking methods.

Table 1 provides an overview of the ranking method, where x_i_ represents the relative methylation value of the *i*-th sample, $\bar{x_{i}}$ is the average value, and σ_xi_ is the sample standard deviation. C denotes the class label, and parameters such as n_1_ and n_2_ demonstrate the number of samples belonging to specific features within the corresponding class label. Additionally, s_w_ and s_b_ correspond to the within-class and between-class scatter matrices, respectively.

### 2.5. Combination of Ranking Methods

The output of each method consists of a ranked list of differentially methylated CpGs across the genome, and the ensemble method is used to merge the output of each ranking method. In this case, the combination is based on a voting system of the actual ranking. If we denote “m” as the number of objects and “n” as the number of preference lists, then “n” represents the number of ranking methods, and “m” signifies the number of CpGs.

For each CpG site, we establish a corresponding rank vector denoted as “r”, where r = (r_1_, …, r_n_) and each r_j_ represents the normalized rank within the range of [0–1] of the CpG site in the j_th_ ranking list. In this voting system, the voters are the ranking functions (n), and the volunteers consist of the complete set of CpG sites (m) in the genome. Each CpG site is assigned specifically allocated scores, which are determined by a ranking method, within different clusters.

In the voting process, if more than 70% of the participants cast their votes in favor of a CpG site with a score exceeding the predefined empirical threshold (set at 0.04), the CpG site qualifies as a candidate for a differentially methylated region (DMR) within the genome. We initially evaluate various threshold values ranging from 0 to 1 using simulation data. This evaluation enables us to consider the trade-off between sensitivity and specificity, ultimately guiding the selection of the most appropriate empirical threshold.

### 2.6. Definition of DMR Boundaries

The outcome of the ranking method combination is a list of differentially methylated CpGs (with a *p*-value ≤ 0.05), from which we infer that they can constitute candidate DMRs. Consequently, we define DMRs as regions comprising significant adjacent CpGs within a single CpG cluster. Subsequently, *p*-values of neighboring CpG sites within a DMR are combined using Fisher’s method in the Metap R package. The start and end position for DMRs are defined when the methylation difference shifts from positive to negative or vice versa. This difference ensures that within a DMR, all CpGs exhibit either hypo- or hypermethylation, respectively. Furthermore, DMRs are ranked based on the Fisher ratio, methylation fold change [case/control], and absolute methylation difference [case–control].

### 2.7. Annotation and Visualization

All DMRs are annotated using the UCSC Genome browser (version hg19) in different classes of genome loci, including CpG islands, shores, and shelves, as well as promoters, gene bodies, transcription start sites, and intergenic regions. Moreover, for each DMR, two statistical criteria like the *p*-value and false discovery rate (FDR) are calculated on the simulation and real datasets. The highly relevant DMRs (*p*-value and FDR ≤ 0.05) for the specific cancer type on the genome browser are illustrated as a heat map in order to assess the DMRs detected in real data analysis in Section 4.2.

## 3. Evaluation Criteria

To assess the sensitivity and specificity of the five DMR finding methods including Methylkit/eDMR [21], BiSeq [23], DSS [25], DMRFusion [44], and HBCR-DMR, we recognized DMRs with a *p*-value less than 0.05 containing five or more CpGs as significant DMRs on the simulation data. Table 2 provides metrics such as true-positive (TP), false-positive (FP), false-negative (FN), true-negative (TN), sensitivity, specificity, accuracy, area under the receiver operator characteristic (ROC) curve (AUC), positive predictive value (PPV), negative predictive value (NPV), Matthews correlation coefficient (MCC), F1 score (F1), and elapsed time to evaluate the performance of these methods to evaluate the performance of these methods and these are described as below:

TP: DMR is characterized as a substantial DMR that coincides with a region resembling simulated DMRs obtained from RRBS data.

FP: DMR is described as a noteworthy DMR that does not intersect with simulated DMRs derived from RRBS data.

FN: DMR is identified as a non-DMR declared by a method that overlaps with simulated DMRs from RRBS data.

TN: DMR is identified as a non-DMR declared by a method that overlaps with simulated non-DMRs from RRBS data.

Sensitivity (recall): The sensitivity refers to the proportion of actual positives (DMRs) that are correctly identified and is estimated as follows:(5)$$Sensitivity=\frac{TP}{TP+FN}$$

Specificity: The specificity refers to the proportion of actual negatives (non DMRs) that are correctly identified and is estimated as follows:(6)$$Specificity=\frac{TN}{TN+FP}$$

Accuracy: The accuracy refers to evaluating the prediction of DMRs that are correctly or incorrectly detected and is estimated as follows:(7)$$Accuracy=\frac{TP+TN}{TN+TP+FN+FP}$$

AUC: A DMR finder method is considered favorable if it provides a high sensitivity to detect DMRs while maintaining a low false positive rate (1 − specificity). We evaluated the trade-off between sensitivity and specificity by calculating ROC curves based on the obtained region-wise *p*-value. The area under the ROC curve is the (AUC) and higher values of the AUC indicate better performance for a method.

(PPV): The PPV refers to is the probability that regions with a DMR detection truly have the methylation changes, using the formula.
(8)$$Positive\;predictive\;value=\frac{TP}{TP+FP}$$

NPV: The NPV refers to is the probability that regions with a non-DMR detection truly do not have the methylation changes, using the formula:(9)$$Negative\;predictive\;value=\frac{TN}{TN+FN}$$

MMC: This is calculated directly from the confusion matrix in order to evaluate binary classifications, using the formula:(10)$$MCC=\frac{TP\times TN-FP\times FN}{\sqrt{\left(TP+FP\right)(TP+FN)(TN+FP)(TN+FN)}}$$

An MCC of +1, 0, and −1 correspond to perfect prediction, no better than random prediction, and total disagreement between predicted and actual status, respectively.

*F*1: This is a weighted average that is calculated directly from the precision and recall, which is estimated as follows:(11)$$F1=2\times \frac{precision\times recall}{precision+recall}$$(12)$$precision=\frac{TP}{TP+FP}$$

The F1 score might be a better measure to use if we need to seek a balance between precision and recall. Furthermore, F1 ranges from 0 (worst prediction) to 1 (perfect precision and recall).

Elapsed time: The elapsed time for each DMR finder method was recorded on a Ubuntu 18.04.03 LTS Operating System, with 32 GB of 2133 MHz, DDR4 RAM, and Intel Core i7 6700 3.4 GHz CPU.

## 4. Results

### 4.1. Simulation Data Analysis

Because the true differential methylation status of CpGs is unknown in real data, simulation data are needed to evaluate the performance of different methods in a situation where the true DMRs are known. So, in order to assess HBCR_DMR’s ability to identify TP DMRs, we used simulation data comparing HBCR_DMR with four other methods: Methylkit/eDMR, BiSeq, DSS, and DMRFusion. Data were generated using the “RRBSdata” package. For all methods, we evaluated different measurements on simulation data that are shown in Table 2.

The results showed that Methylkit/eDMR, BiSeq, DSS, DMRFusion, and HBCR_DMR identified 1249, 6484, 4718, 6271, and 7111 significant DMRs, respectively. Table 2 indicates that the HBCR_DMR method has more TP DMRs than the other methods. Considering sensitivity and specificity, HBCR_DMR and BiSeq outperform the other methods. The specificity values for Methylkit/eDMR and DSS are both 0.99, while their sensitivity values are relatively poor. The accuracy measures for BiSeq and HBCR_DMR are 0.85 and 0.82, respectively, while DSS, DMRFusion, and Methylkit/eDMR have accuracy values of 0.78, 0.67, and 0.64, respectively. Thus, the highest accuracy values are achieved by BiSeq and HBCR_DMR. In terms of the AUC and F1 score as performance metrics, HBCR_DMR achieves an AUC of 0.94 and an F1 score of 0.76, which is second only to BiSeq, which has an AUC of 0.97 and an F1 score of 0.78.

The maximum value of PPV is 0.99, as measured in BiSeq, Methylkit/eDMR, and DSS. However, HBCR_DMR has an NPV value of 0.83, which is the highest value among the five methods. The highest MCC values are 0.71 and 0.62, calculated for the BiSeq and HBCR_DMR methods, respectively. The results of elapsed time for chromosome 1 were calculated for each method. The elapsed times (in seconds) for Methylkit/eDMR, BiSeq, DSS, DMRFusion, and HBCR_DMR are 41, 20,072, 196, 1740, and 1347, respectively. As a result, the elapsed time for HBCR_DMR is 14 times faster than that of BiSeq (Table 2).

We evaluated the performance of all methods using the AUC. Figure 2 displays the ROC curves. The sensitivity and specificity for BiSeq are 0.65 and 0.99, respectively, and for HBCR_DMR, they are 0.72 and 0.89, respectively. BiSeq has a higher specificity value than HBCR_DMR, while our proposed method exhibits higher sensitivity compared to BiSeq. Therefore, higher AUC values are assigned to BiSeq and HBCR_DMR.

Figure 3 illustrates the overlap of significant DMRs identified by the five methods. The overlap of the detected DMRs from different methods with the TP DMRs in the simulation data is visualized using Venn diagrams. We utilized the “makeVennDiagram” function in the ChIPpeakAnno R package to create the Venn diagrams. HBCR_DMR detects more TP DMRs than the other methods. Specifically, this method identifies 82%, 33%, 11%, and 8% more TP DMRs than Methylkit/eDMR, DSS, DMRFusion, and BiSeq, respectively. The number of common TP DMRs among all methods is 757.

### 4.2. Real Data Analysis

We applied our proposed method to the SureSelectXT Human Methyl-Seq dataset from our previous study on colorectal cancer and normal colon tissue [45]. This dataset comprises six colorectal adenocarcinoma and six control samples. In the current study, our aim was to compare the results of our HBCR_DMR method with those of other methods. By comparing CRC and normal samples in multi-samples, we detected several thousand hyper- and hypomethylation DMRs. In total, we identified 7325 hyper DMRs and 10,879 hypo DMRs, each with a length of more than 200 bp, the highest Fisher ratio score between these two groups, and *p*-values and an FDR less than 0.05 (Appendix A). Furthermore, we performed the conversion of genomic coordinates for hyper- and hypomethylation DMRs from the hg19 reference genome to the hg38 reference genome in (Appendix A).

Figure 4A,C reveal that the majority of identified DMRs in both the hypermethylation and hypomethylation categories are predominantly situated within intergenic regions, accounting for 89% and 90%, respectively. In the case of hyper DMRs, 67% of them are annotated within CpG islands, whereas only 21% of the hypomethylated DMRs are found in these regions. Notably, a significant portion of hyper DMRs is located in CGI shores, followed by exons, promoters, and CGI shelves, in that order.

Given that the detected DMRs span more than 200 base pairs, some of them extend across multiple genomic regions, encompassing multiple annotation features. Figure 4B,D provide a visual representation of the expanded annotations for the detected hypermethylation and hypomethylation DMRs. Importantly, a substantial portion of DMRs initially located in intergenic regions extends into intronic regions in both the hypermethylation and hypomethylation categories, as illustrated in Figure 4.

We assessed DNA methylation changes based on DMRs among six CRC samples (T20, T45, T67, T31, T65, T35) and six control samples (N4, N7, N8, N10, N14, N16). In Figure 5, the hyper- and hypomethylation DMR regions are depicted in a heatmap at an FDR of 0.01. A distinct pattern of DNA methylation changes is evident between the CRC and control samples.

From a biological perspective, hypermethylation regions play a key role in the occurrence of CRC. Our candidate DMRs are selected based on the top hypermethylation DMRs, with a methylation fold change > 20 and an absolute methylation difference > 0.1. Table 3 presents the top five DMRs, located in the SFMBT2, SOX5, ZNF43, AGBL4, and SOX5 genes. Furthermore, their annotation details and a visualization of the methylation changes in the DMRs with the highest difference in methylation between the CRC and control samples are provided in Appendix A, respectively. Additionally, the average methylation fold change and absolute methylation difference between the CRC and control samples in these candidate regions are 25.08 and 0.24, respectively.

Table 4 compares our proposed approach with previous tools. The number of significant regions and the Type I error rate with a *p*-value and FDR < 0.05 are as follows: 6944 (0.3), 5637 (0.056), 18,065 (0.065), 15,362 (0.042), and 18,204 (0.028) for the Methylkit/eDMR, BiSeq, DSS, DMRFusion, and HBCR_DMR methods, respectively. Thus, the Type I error rate for BiSeq, DMRFusion, and HBCR_DMR is lower than or approximately 0.05.

## 5. Discussion

DNA methylation has an important role in carcinogenesis [50]. Thus, CpG regions with different methylation levels, known as DMRs, are of great importance. Statistical analysis of genome-wide bisulfite sequencing with multiple biological samples is challenging due to heterogeneous read coverage, varying methylation levels, a relatively small sample size, and a large number of CpGs in the genome.

Here, we have developed a novel DMR detection tool based on a hybrid of a beta-binomial hierarchical model and a combination of ranking methods. The major advantages of our method are as follows: First, it is suitable for small sample sizes in DNA methylation sequencing. Second, it takes into account biological variability, sampling variability, and variation across the methylation proportion of the CpG sites. Finally, it considers the diversity among ranking methods, different outputs, and result stability.

In the present study, we compared four popular DMR analysis methods, namely Methylkit/eDMR, BiSeq, DSS, and DMRFusion, with HBCR-DMR using simulation data and a real methylation dataset. We evaluated the performance of these methods based on various metrics, including TP, FP, FN, TN, sensitivity, specificity, accuracy, AUCs, PPV, NPV, MMC, F1, and elapsed time, using simulation data.

According to our simulation results, HBCR_DMRs outperform other methods in their ability to identify TP DMRs. The BiSeq method excels in terms of specificity, accuracy, AUCs, PPV, MMC, and F1. However, it has fewer TP DMRs compared to HBCR_DMR, and it takes 14 times longer in terms of elapsed time compared to our proposed method.

Regarding specificity, Methylkit/eDMR, BiSeq, and DSS perform well, while the sensitivity of Methylkit/eDMR and DSS is less satisfactory. We observed a trade-off between sensitivity and specificity, along with a reasonable elapsed time, and the highest value of NPV measure in HBCR_DMR.

The proposed method identifies 18,204 differentiated methylation regions with a Type I error rate of 0.028. These regions are divided into two groups: hypermethylation (7325 regions) and hypomethylation (10,879 regions) based on real data. Among the 11,336 significantly detected DMRs (with a *p*-value < 0.01), 2205 known genes are identified. The Type I error rates for the hypermethylation and hypomethylation regions are 0.01 and 0.04, respectively (see Appendix A). Thus, the method can detect a large number of DMRs with an approximate Type I error of 0.028.

We recommend the use of HBCR_DMR, BiSeq, or DMRFusion methods, which performed for a wide range of DMRs based on simulation and real dataset results. Notably, HBCR_DMR exhibits superior efficacy in identifying TP DMRs when compared to alternative methods, concurrently exhibiting the lowest Type I error rate within this study.

## 6. Conclusions

The HBCR_DMR method comprises a hybrid approach, merging a beta-binomial Bayesian hierarchical model with a combination of ranking techniques. This method proves invaluable as a DMR discovery tool and is particularly suitable for situations involving a limited sample size of DNA methylation sequencing. It adeptly incorporates considerations of biological variability, sampling variability, and the inherent variation within CpG site methylation proportions. Moreover, it thoughtfully addresses the disparities among ranking methods, accommodating divergent outputs while maintaining result stability. Notably, the HBCR_DMR method demonstrates a heightened capacity for identifying TP DMRs compared to alternative approaches, concurrently exhibiting the lowest Type I error rate within its category. In addition, HBCR_DMR exhibits versatility across various methylation sequencing platforms, including WGBS, RRBS, SureSelectXT, Human Methyl-Seq, and target-capture methods.

## Figures and Tables

**Figure 1 jpm-14-00361-f001:**
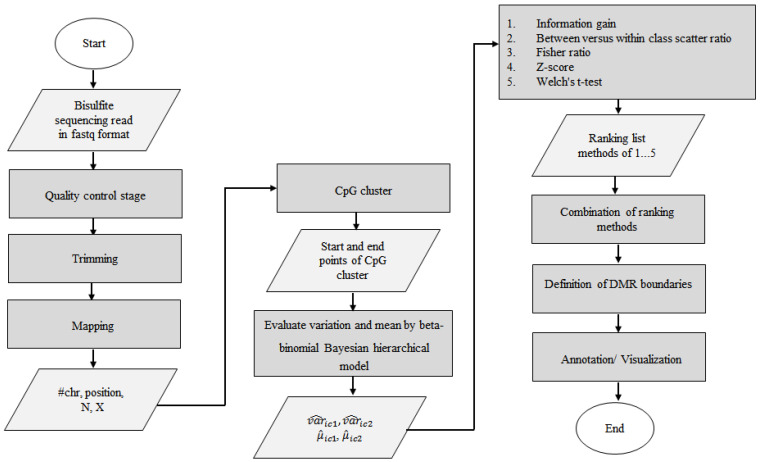
The flowchart of the proposed approach.

**Figure 2 jpm-14-00361-f002:**
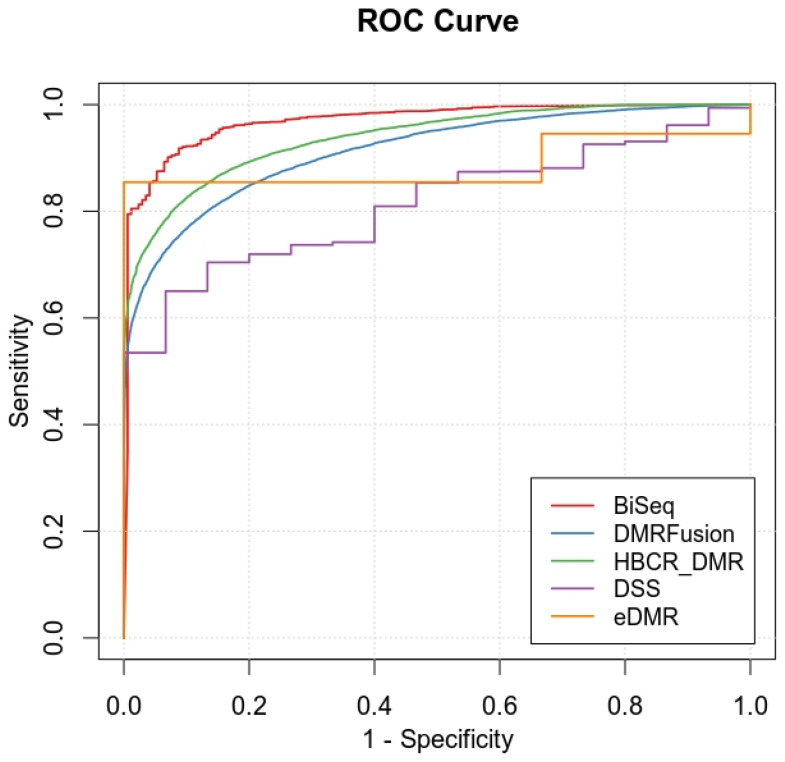
ROC curves compare sensitivity and specificity of Methylkit/eDMR, BiSeq, DSS, DMRFusion, and HBCR_DMR methods on simulation dataset.

**Figure 3 jpm-14-00361-f003:**
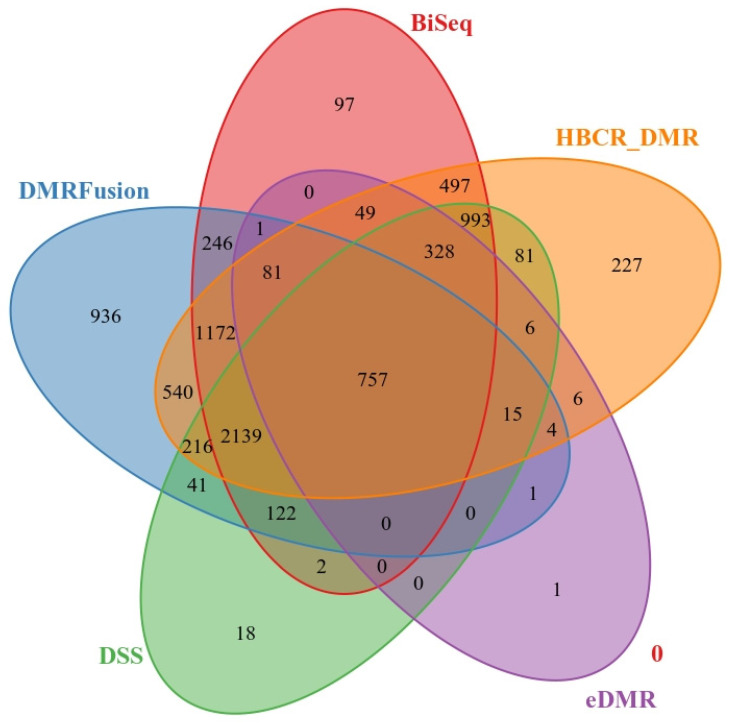
Overlap of TP DMRs for Methylkit/eDMR, BiSeq, DSS, DMRFusion, and HBCR_DMR methods.

**Figure 4 jpm-14-00361-f004:**
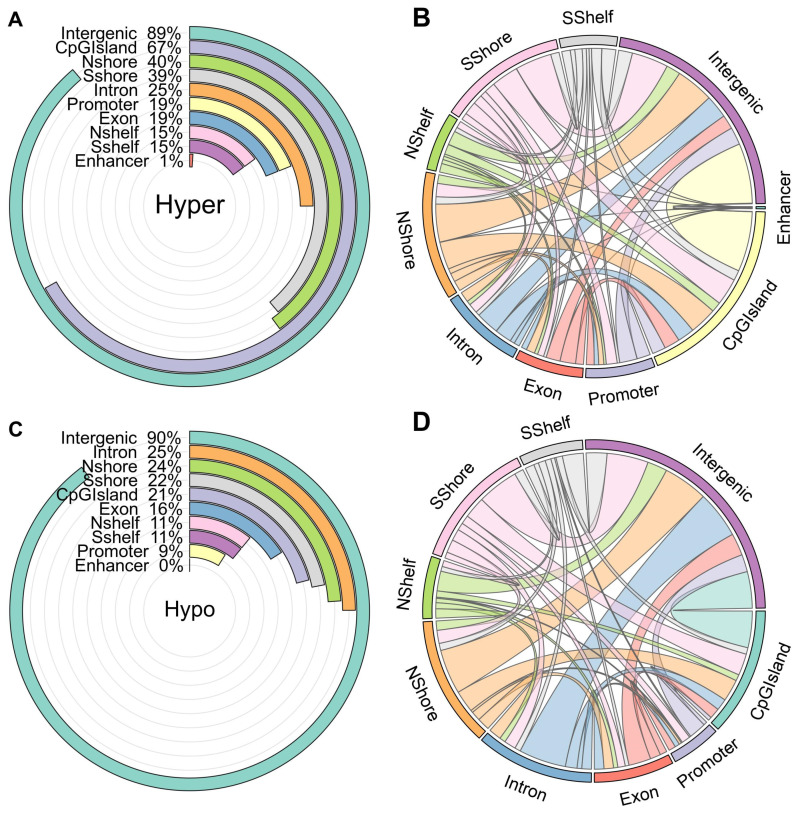
Statistical information of hyper/hypomethylation DMR annotation. (**A**) Hypermethylation DMRs, (**B**) Expansion of the detected hyper DMRs, (**C**) Hypomethylation DMRs, (**D**) Expansion of the detected hypo DMRs.

**Figure 5 jpm-14-00361-f005:**
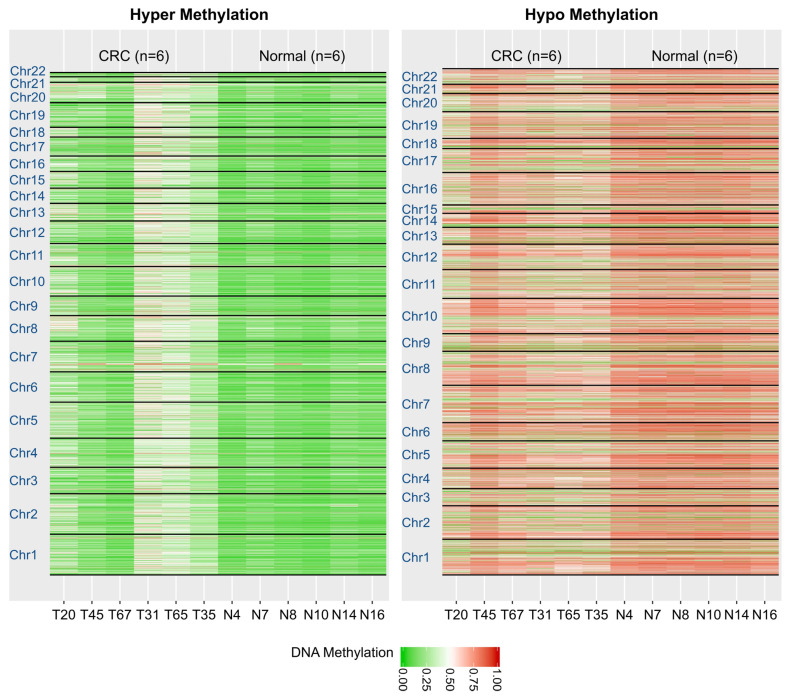
Heat map representation of DNA Human Methyl-seq data between CRC and normal samples on whole genes. Each column represents one sample and each row represents CpGs methylation status in hyper/hypomethylation DMRs identified by HBCR_DMR.

**Table 1 jpm-14-00361-t001:** Ranking methods.

Ranking Method	Criterion
Information gain	$S=Info\left(X\right)-{Info}_{x}\left(X\right)$ $Info\left(X\right)=-\sum_{i=1}^{k}P\left(c_{i}\times X\right)\times \log\left(P\left(c_{i}\times X\right)\right)$ ${Info}_{x}\left(X\right)=-\sum_{i=1}^{V}\frac{\left|V_{i}\right|}{\left|X\right|}\times Info\left(V_{i}\right)$
K = Number of classes V = Number of individual values of a CpGs x $V_{i}$ = The set of instances whose values in CpGs x equal $x_{i}$ $\left|V_{i}\right|$ = Number of samples in $\left|V_{i}\right|$ $\left|X\right|=n$ P = Probability density function $c_{i}$ = Label corresponding (i = 1… k)	
Between versus Within Class Scatter Ratio	$s_{w}=\sum_{i=1}^{c}\sum_{j}^{n_{c}}\left(x_{j}-\mu_{i}\right)*{\left(\left(x_{j}-\mu_{i}\right)\right)}^{T}$ $s_{b}=\sum_{i=1}^{c}\left(\mu_{i}-\mu \right)*{\left(\left(\mu_{i}-\mu \right)\right)}^{T}$ $S=\frac{s_{b}}{s_{w}}$
$x_{j}$—Relative methylation values of a CpGs x in jth sample $\mu_{i}$—Average value of relative methylation in a CpGs x across all samples in ith class (i = 1…c) μ = Total average value of relative methylation in a CpGs x across all classes (i = 1…c) T = Transpose matrix C = class label feature $s_{w}$ = Within classes scatter $s_{b}$ = Between classes scatter $n_{c}$ = Number of samples in ith class (i = 1…c)	
Fisher ratio	$FR\left(x\right)=\frac{{\left(\left[{\bar{x}}_{c_{1}}-{\bar{x}}_{c_{2}}\right]\right)}^{2}}{{\sigma_{x_{c_{1}}}}^{2}+{\sigma_{x_{c_{2}}}}^{2}}$
${\bar{x}}_{c_{1}}$,${\bar{x}}_{c_{2}}$ = Mean value of relative methylation in a CpGs x across all samples in ith class (i = 1, 2) $\sigma_{x_{c_{i}}}$ = Standard deviation value of relative methylation in a CpGs x across all samples in ith class (i = 1, 2)	
Z-score	$S=\frac{{\bar{x}}_{c_{1}}-{\bar{x}}_{c_{2}}}{\sigma_{x}}$
$\sigma_{x}$ = Standard deviation value of relative methylation in a CpGs x across all samples in both classes	
Welch’s *t*-test	S=x¯c1−x¯c2σxc1nc1+σxc2nc2
$n_{c_{i}}$ = Number of samples in ith class (i = 1, 2)	

**Table 2 jpm-14-00361-t002:** TP, FP, FN, TN, Sensitivity, Specificity, Accuracy, AUC, PPV, NPV, MCC, F1, and Elapsed time (in seconds) for the different DMR detection tools based on simulation datasets.

Method	TP	FP	FN	TN	Sensitivity ^1^	Specificity	Accuracy	AUC	PPV ^2^	NPV ^3^	MCC ^4^	F1 Score	Time ^5^ (Second)
Methylkit/eDMR	1249	3	8751	14,695	0.12	0.99	0.64	0.88	0.99	0.63	0.28	0.22	41
BiSeq	6484	89	3493	14,632	0.65	0.99	0.85	0.97	0.99	0.81	0.71	0.78	20,072
DSS	4718	15	5279	14,686	0.48	0.99	0.78	0.81	0.99	0.73	0.58	0.64	196
DMRFusion	6271	4495	3554	10,378	0.64	0.69	0.67	0.91	0.58	0.74	0.33	0.61	1740
HBCR_DMR	7111	1674	2759	13,154	0.72	0.89	0.82	0.94	0.81	0.83	0.62	0.76	1347

^1^ Sensitivity = Recall. ^2^ PPV: Postive predictive value. ^3^ NPV: Negative predictive value. ^4^ MCC: Matthews correlation coefficient. ^5^ Time (Second): Elapsed time is calcalated for Chr 1.

**Table 3 jpm-14-00361-t003:** Information of the highest difference in methylation for hypermethylation DMRs.

Chr	[Start-End]	Gene Symbol	Function	Fisher Ratio	Fold Change	Absolute Methylation Difference	*p* Value	Q Value
10	7452243-7452499	*SFMBT2*	*SFMBT2* gene is sequence-specific DNA binding, histone binding and miRNA interaction protein [46].	1.77	38.24	0.28	1.09 × 10^−14^	6.11 × 10^−14^
12	24715833-24716098	*SOX5*	SOX5 gene is an unbalances tumor microenvironment to regulate colorectal cancer progression [47].	0.71	22.8	0.22	1.04 × 10^−7^	3.28 × 10^−7^
19	22034731-22034990	*ZNF43*	The zinc finger protein43 are involved in gene regulation and development [48].	2.1	21.5	0.31	1.47 × 10^−13^	8.93 × 10^−13^
1	49242758-49243000	*AGBL4*	*AGBL4* gene related to tubulin binding and metallocarboxypeptidase activity [49].	0.81	21.44	0.19	3.92 × 10^−5^	7.84 × 10^−5^
12	24715169-24715370	*SOX5*	*SOX5* gene is an unbalances tumor microenvironment to regulate colorectal cancer progression [47].	0.75	21.42	0.19	6.91 × 10^−8^	2.24 × 10^−7^

**Table 4 jpm-14-00361-t004:** Comparison of the number of significant regions with a *p*-value and FDR < 0.05 and Type I error rate between previous tools and HBCR-DMR for DNA Human Methyl-seq data between CRC and normal samples.

Method	Number of DMRs	Type 1 Error Total
Methylkit/eDMR	6944	0.3
BiSeq	5637	0.056
DSS	18,065	0.065
DMRFusion	15,362	0.042
HBCR_DMR	18,204	0.028

## Data Availability

The data presented in this study are available on request from the corresponding author.

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
