# Peer review of "HBCR_DMR: A Hybrid Method Based on Beta-Binomial Bayesian Hierarchical Model and Combination of Ranking Method to Detect Differential Methylation Regions in Bisulfite Sequencing Data"

_jpm, 2024, doi:10.3390/jpm14040361_

Round 1

Reviewer 1 Report

Comments and Suggestions for Authors

The authors herein present an approach for detecting DMR's in sequencing data. The authors describe their tool as utilizing a hybrid of a beta-binomial hierarchical model and a combination of ranking methods for calling DMRs. To test their algorithm, the authors have benchmarked their approach against other published approaches with mixed results. 

I have a few critiques of the presented paper which focus around (in no particular order)
1 - the authors have chosen to work with genome build hg19. This is an incredibly old genome build, especially with the recent release of T2T-CHM13v2.0, the 1st gapless genome build which signals an end of life for even hg38. At the very least, the authors need to update their work to be hg38 data and compatible.

2 - The layout of the manuscript can be improved. The authors reference figure 1, then figure 5, and then proceed to reference the remaining figures. I suggest adjusting the flow to follow a more standardized presentation format.

3 - Section 3.2 is difficult to understand and I could not understand the point of what the authors were trying to get at. Please clarify further.

4 - The benchmarking metrics of HBCR_DMR are not stellar. I understand when developing algorithms the metrics are what they are, but in this case, the authors do not make a compelling case for utilizing HBCR_DMR over the benchmarked approaches. I fail to see why an end user would use HBCR_DMR over BiSeq or others, other than the fact that BiSeq just takes sooo long to run in comparison. The differences in TP's as reported in this study are not drastic enough for me to consider this as a true benefit. Unless the authors can provide additional evidence as to why/how HBCR_DMR is preferential to other approaches presented, I do not see value to yet another tool for reporting nearly the same findings as already published tools, even if it does so using a different approach. 

Comments on the Quality of English Language

There are segments of the manuscript which would benefit from additional proof-reading and improvements.

Author Response

Dear Editor and Reviewers,

We sincerely appreciate your letter and the valuable feedback received from the reviewers regarding our manuscript, titled " HBCR_DMR: A Hybrid Method Based on Beta-Binomial Bayesian Hierarchical Model and Combination of Ranking Method to Detect Differential Methylation Regions in Bisulfite Sequencing Data” which was submitted to Journal of Personalized Medicine. We have diligently reviewed and meticulously revised the manuscript to address all the comments provided by the reviewers. In this submission, we are including the revised manuscript with highlighted modifications, along with a detailed point-by-point response document. We extend our heartfelt gratitude to you and the reviewers for their constructive and positive comments. We feel that these revisions have significantly enhanced the quality of our manuscript.

Reviewer 1:

Comments to the author (in black), our responses; in blue.

The authors herein present an approach for detecting DMR's in sequencing data. The authors describe their tool as utilizing a hybrid of a beta-binomial hierarchical model and a combination of ranking methods for calling DMRs. To test their algorithm, the authors have benchmarked their approach against other published approaches with mixed results. 

I have a few critiques of the presented paper which focus around (in no particular order)
1 - the authors have chosen to work with genome build hg19. This is an incredibly old genome build, especially with the recent release of T2T-CHM13v2.0, the 1st gapless genome build which signals an end of life for even hg38. At the very least, the authors need to update their work to be hg38 data and compatible.

Response:

We appreciate the insightful comments provided by the reviewer. In response, we have performed the conversion of genomic coordinates for Differentially Methylated Regions (DMRs) in (Supplementary materials File. S2), comprising 7325 hypermethylated DMRs and 10879 hypomethylated DMRs, all with a length exceeding 200 base pairs. This conversion entailed transitioning from the hg19 reference genome to the hg38 reference genome in (Supplementary materials File. S3).

2 - The layout of the manuscript can be improved. The authors reference figure 1, then figure 5, and then proceed to reference the remaining figures. I suggest adjusting the flow to follow a more standardized presentation format.

Response:

In the revised manuscript, we reference figures based on the details provided below:

Method (Section 2): Page 4 - Figure 1

Simulation Data Analysis (Section 4.1): Page 10 - Figure 2 and Figure 3

Real Data Analysis (Section 4.2): Page 11 - Figure 4 and Figure 5

3 - Section 3.2 is difficult to understand and I could not understand the point of what the authors were trying to get at. Please clarify further.

Response:

CpG clusters are DNA regions abundant in cytosine-phosphate-guanine (CpG) dinucleotides. To map DNA methylation, several techniques are available, including WGBS, RRBS, and SureSelectXT Human Methyl-Seq. WGBS stands as the most advanced method, providing a detailed, nucleotide-level view of the entire methylome. On the other hand, RRBS and SureSelectXT Human Methyl-Seq are based on Enzyme-seq and target specific genome regions. Before employing computational analysis, we establish CpG clusters, which are CpG-rich regions within the genome. These clusters encompass CpG sites that are present in over 75% of all samples. This approach effectively removes noise from DNA methylation data across the genome and reduces the computational processing time, as we only need to focus on specific genome regions rather than the entire genome.

4 - The benchmarking metrics of HBCR_DMR are not stellar. I understand when developing algorithms, the metrics are what they are, but in this case, the authors do not make a compelling case for utilizing HBCR_DMR over the benchmarked approaches. I fail to see why an end user would use HBCR_DMR over BiSeq or others, other than the fact that BiSeq just takes sooo long to run in comparison. The differences in TP's as reported in this study are not drastic enough for me to consider this as a true benefit. Unless the authors can provide additional evidence as to why/how HBCR_DMR is preferential to other approaches presented, I do not see value to yet another tool for reporting nearly the same findings as already published tools, even if it does so using a different approach. 

Response:

Thank you for your feedback and appreciate the opportunity to address the concerns raised and provide further context for the significance of our DMR detection tool. First and foremost, we would like to summarize the key findings from our study. In our comparative analysis, HBCR_DMR demonstrated a sensitivity of 72% and an accuracy of 82%, while the benchmarked Biseq tool exhibited a sensitivity of 65% and an accuracy of 85%. Additionally, we highlighted the significantly improved speed of HBCR_DMR in comparison to Biseq, which can be of practical importance to researchers. In response to the reviewer's concern about our benchmarking metrics, it is essential to emphasize the importance of sensitivity in DMR detection. Sensitivity plays a pivotal role in ensuring that true positives are accurately identified. In the field of epigenetic research, where even subtle changes can have profound implications, the ability to capture these changes effectively is paramount. Higher sensitivity is, therefore, a critical attribute of a DMR detection tool. Furthermore, HBCR_DMR, is designed to be robust and adaptable to various biological scenarios. The innovative feature selection approaches we employ enhance the tool's capacity to discern underlying patterns in DNA methylation data. This robustness is a key factor in the preference for HBCR_DMR over other approaches. The reviewer's concerns regarding the utility of HBCR_DMR over Biseq are well-founded. Recent studies (Han, Tang et al. 2018, Liu, Han et al. 2020) have highlighted limitations in Biseq, particularly in cases of small sample sizes and data with high dispersion. These studies underscore the need for alternative tools that can deliver more reliable results in these scenarios. One of the practical advantages of HBCR_DMR is its efficiency. It operates significantly faster than Biseq, making it a valuable choice for researchers dealing with large datasets or those who need to conduct multiple analyses simultaneously. Reduced processing time ensures prompt results, enabling researchers to make timely decisions and explore a broader spectrum of possibilities. HBCR_DMR represents a substantial advancement over our prior method, which relied on a Normal assumption for feature selection. The change in variance estimation have significantly enhanced the performance and accuracy of our tool.

Reviewer 2: 

-Introduction. The first part of the chapter is well written and understandable. Starting form page 3, where you talk about differential methylation finder methods, the structure is no longer correct. Since it is an "Introduction", I think the content is too technical. It should be moved to a "Materials & Methods" section.  The description of the informatics approaches is technical and not suitable for an introduction.

Response:

We appreciate the comments suggested by the reviewer. It's important to highlight that this section of the introduction provides a review of the prior literature on computational methods in the field of DNA methylation analysis. Typically, such a description is presented in the introductory segment preceding the materials and methods section.

- The chapters "Approach" and "Method" should be combined together, in my opinion.

Response:

In the revised manuscript, the approach part with method part is combined.

The work is limited by the fact that the datasets used are contained in a "R" package and do not arise from original datasets collected by the authors. I would specify which datasets of the "RRBSdata" R package have been used for the modelling. The results would benefit of a better explanation of the datasets used for the comparison.  

Response:

The results section of our study is divided into two parts: simulation data analysis and real data analysis. In the simulation data analysis, we utilized data generated using the 'RRBSdata' package and applied and compared four methods: Methylkit/eDMR, BiSeq, DSS, DMRFusion, and HBCR_DMRs to this dataset. For the real data analysis, we applied our proposed method to the SureSelectXT Human Methyl-Seq dataset obtained from our previous study on colorectal cancer and normal colon tissue. Therefore, we conducted an assessment of our proposed method in comparison to other methods using both simulated and real DNA methylation sequencing data.

Reference:

Han, C., H. Tang, S. Lou, Y. Gao, M. H. Cho and S. Lin (2018). "Evaluation of recent statistical methods for detecting differential methylation using BS-seq data." OBM Genetics 2(4): 1-36.

Liu, Y., Y. Han, L. Zhou, X. Pan, X. Sun, Y. Liu, M. Liang, J. Qin, Y. Lu and P. Liu (2020). "A comprehensive evaluation of computational tools to identify differential methylation regions using RRBS data." Genomics 112(6): 4567-4576.

Reviewer 2 Report

Comments and Suggestions for Authors

Dear Authors,

please, address the following issues:

- Introduction. The first part of the chapter is well written and understandable. Starting form page 3, where you talk about differential methylation finder methods, the structure is no longer correct. Since it is an "Introduction", I think the content is too technical. It should be moved to a "Materials & Methods" section.  The description of the informatics approaches is technical and not suitable for an introduction.

- The chapters "Approach" and "Method" should be combined together, in my opinion.

The work is limited by the fact that the datasets used are contained in a "R" package and do not arise from original datasets collected by the authors. I would specify wich datasets of the "RRBSdata" R package have been used for the modeling.

The results would benefit of a better explanation of the datasets used for the comparison.  

Comments on the Quality of English Language

English is acceptable.

Author Response

(The authors gave the same response as above.)

Round 2

Reviewer 2 Report

Comments and Suggestions for Authors

Dear Authors,

I read the improvements you made to the manuscript.

Comments on the Quality of English Language

English is fine.